# ECCO: Evidence-Driven Causal Reasoning for Compiler Optimization

**Haolin Pan** [1 2 3]  **Lianghong Huang** [1 2]  **Jinyuan Dong** [1 2]  **Mingjie Xing** [1 2 3]  **Yanjun Wu** [1 2]

## Abstract

Compiler auto-tuning faces a dichotomy between traditional black-box search methods, which lack semantic guidance, and recent Large Language Model (LLM) approaches, which often suffer from superficial pattern matching and causal opacity. In this paper, we introduce ECCO, a framework that bridges interpretable reasoning with combinatorial search. We first propose a reverse engineering methodology to construct a Chain-of-Thought dataset, explicitly mapping static code features to verifiable performance evidence. This enables the model to learn the causal logic governing optimization decisions rather than merely imitating sequences. Leveraging this interpretable prior, we design a collaborative inference mechanism where the LLM functions as a strategist, defining optimization intents that dynamically guide the mutation operations of a genetic algorithm. Experimental results on seven datasets demonstrate that ECCO outperforms the LLVM `opt -O3` baseline, achieving an average 24.44% reduction in cycles.

## 1. Introduction

Compiler optimization serves as the critical bridge between high-level abstractions and hardware-specific execution. The task of selecting and ordering optimization passes to minimize latency or code size—known as the phase-ordering problem—is a combinatorial challenge characterized by a vast, non-convex search space, as illustrated in Fig. 1 (Ashouri et al., 2018). Traditional heuristics, such as the fixed -O3 sequence in LLVM (Lattner & Adve, 2004), often fail to exploit the specific characteristics of diverse workloads. Consequently, auto-tuning has evolved from iterative compilation to machine learning approaches, including

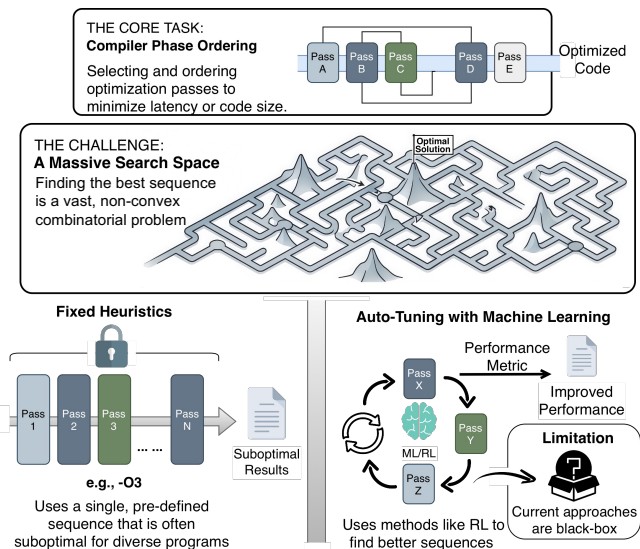

*Figure 1.* Overview of the compiler phase-ordering task.

Bayesian optimization (Chen et al., 2021) and Reinforcement Learning (RL) (Haj-Ali et al., 2020). However, these methods typically treat the compiler as a black box, optimizing an objective function without modeling the underlying semantics of program transformations.

The emergence of LLMs offers new avenues for code intelligence (Cummins et al., 2025; Tang et al., 2025; Grubisic et al., 2024), yet current LLM-based optimization approaches face two critical limitations. **(1) Causal Opacity:** Most models are trained via supervised fine-tuning on simple code-sequence pairs. This encourages superficial pattern matching, where the model correlates source code with optimization flags but fails to grasp the causal mechanism—specifically, how a pass alters the code structure to yield performance gains. **(2) Structural Disconnect:** Generative models excel at high-level semantic planning but struggle with the precise combinatorial exploration required for valid compilation sequences. Conversely, traditional search algorithms are effective at local exploitation but lack the global semantic guidance to navigate the search space.

To address these challenges, we introduce ECCO, a framework that shifts the paradigm from mimetic imitation to interpretable, evidence-based reasoning. Our approach is grounded in the principle that an effective optimizer

[1]Institute of Software at Chinese Academy of Sciences [2]University of Chinese Academy of Sciences [3]Hangzhou Institute for Advanced Study at University of Chinese Academy of Sciences. Correspondence to: Mingjie Xing <mingjie@iscas.ac.cn>.

*Proceedings of the 43[rd] International Conference on Machine Learning*, Seoul, South Korea. PMLR 306, 2026. Copyright 2026 by the author(s).

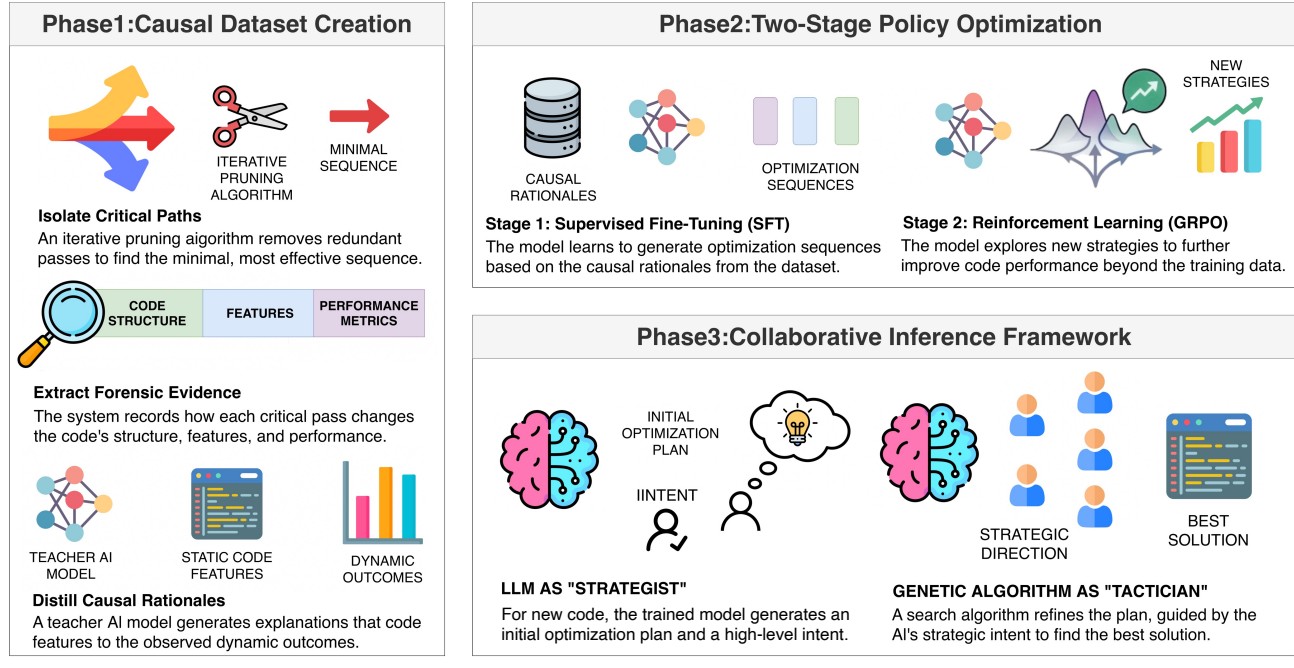

*Figure 2.* **Overview of the ECCO.** The pipeline consists of three distinct phases: (1) **Causal Dataset Creation**, where raw optimization trajectories are pruned and reverse-engineered into evidence-rich rationales; (2) **Two-Stage Policy Optimization**, which aligns the model via SFT and enhances exploration via GRPO; and (3) **Collaborative Inference Framework**, utilizing a Strategist-Tactician paradigm where the LLM's semantic intent directs the local search of a Genetic Algorithm.

must understand the causal chain connecting static features, structural transformations, and performance improvements. ECCO implements a forensic reconstruction mechanism: it prunes optimal sequences to isolate critical paths and extracts multi-modal evidence to construct a causal training corpus. Finally, we propose a collaborative inference method that explicitly decouples semantic intent from combinatorial execution, where the LLM functions as a strategist, defining optimization intents that dynamically guide the mutation operations of a Genetic Algorithm(GA).

We evaluated ECCO on a comprehensive suite of seven standard benchmark sets. Experimental results demonstrate that our framework consistently achieves an average cycle reduction of **24.44%** compared to the LLVM `opt -O3` baseline. Our specific contributions are as follows:

- **Evidence-Driven Causal Paradigm:** We propose a reverse-engineering approach to construct a Chain-of-Thought dataset by pruning raw trajectories and extracting verifiable evidence, encouraging *simulated predictive reasoning* beyond surface-level sequence imitation.

- **Collaborative Strategist–Tactician Framework:** We present a collaborative inference framework where LLM-derived optimization intents guide the mutation operators of a Genetic Algorithm, facilitating coordination between global semantic planning and local

combinatorial search.

- **Dual Causal Datasets and Empirical Excellence:** We contribute two novel datasets to the community: a *forensic evidence dataset* that maps optimization passes to verifiable IR feature deltas, and a *chain-of-thought reasoning dataset* for training interpretable policies.

## 2. Related Work

**Traditional and ML-based Auto-tuning.** To address the NP-hard problem, compiler auto-tuning has evolved from iterative compilation (Bodin et al., 1998) to data-driven approaches. Early methods employed heuristic search algorithms, such as GA (Garciarena & Santana, 2016) and simulated annealing (Ashouri et al., 2018), to explore the vast optimization space (Pan et al., 2025e;d;a). To reduce search latency and enhance generalization across diverse programs, Machine Learning (ML) techniques were introduced (Leather & Cummins, 2020; Park et al., 2022; Zhu et al., 2024; 2025; Pan et al., 2025c; Deng et al., 2024; Chen et al., 2021). Predictive models and RL frameworks (Haj-Ali et al., 2020), exemplified by coresetNVP (Liang et al., 2023), learn to map extracted program features to optimal pass sequences. While these methods effectively prune the search space, they typically operate as black boxes, optimizing scalar performance metrics without modeling the

underlying semantics of code transformations.

**LLM for Code Optimization.** LLMs have recently been applied to compilation tasks, leveraging their semantic reasoning capabilities (Tang et al., 2025). Approaches range from direct prediction of optimization flags to rewriting Intermediate Representation (IR) (Cummins et al., 2025). To mitigate the hallucination of invalid sequences, recent studies have incorporated compiler feedback loops, where the model iteratively refines its output based on execution metrics or error logs (Cummins et al., 2024). Furthermore, agent-based frameworks have emerged, treating optimization as a multi-step planning task (Pan et al., 2025b; Lin et al., 2025). Despite these advancements, most LLM-based methods rely on surface-level pattern matching or trial-and-error, lacking the causal interpretability required to explain the logic behind specific optimization passes.

## 3. Methodology

In this section, we present ECCO, a method bridging semantic reasoning and combinatorial search. As illustrated in Figure 2, the system operates through three interconnected phases. **Phase 1 (Causal Dataset Creation)** employs a reverse engineering approach to isolate critical optimization paths and distill multi-modal evidence into causal rationales. **Phase 2 (Two-Stage Policy Optimization)** aligns the model via Supervised Fine-Tuning (SFT) before enhancing exploration through GRPO. Finally, **Phase 3 (Collaborative Inference)** adopts a *Strategist-Tactician* paradigm, where the LLM's high-level intent dynamically guides the combinatorial refinement of a GA.

### 3.1. Causal Dataset Creation

To transition from black-box sequence prediction to interpretable optimization, we developed a rigorous data collection pipeline. This pipeline transforms raw code-sequence pairs into evidence-annotated samples, shifting the learning objective from simple imitation to the reconstruction of causal optimization logic.

#### 3.1.1. ISOLATION OF CRITICAL OPTIMIZATION PATHS

We begin by acquiring a diverse set of high-performance optimization sequences using the CFSAT (Pan et al., 2025c) framework. While CFSAT effectively explores the search space, the resulting sequences often contain redundant passes that contribute marginally to performance. To isolate the causal drivers of efficiency, we implement an Iterative Greedy Pruning algorithm (Algorithm 1).

As formalized in Algorithm 1, the process iteratively attempts to remove each pass $p_i$, accepting the removal if the cycle count $C'$ remains equal to or lower than the current best $C_{best}$. This filtering ensures the dataset contains

---

**Algorithm 1** Iterative Greedy Sequence Pruning

1: **Input:** Initial Sequence $S = [p_1, \ldots, p_n]$, Program $P$
2: **Output:** Minimal Sequence $S_{min}$
3: $C_{best} \leftarrow$ GetCycles$(P, S)$
4: $changed \leftarrow$ **true**
5: **while** $changed$ **do**
6:     $changed \leftarrow$ **false**
7:     **for** $i \leftarrow 0$ to $|S| - 1$ **do**
8:         $S' \leftarrow S \setminus \{p_i\}$ {Tentatively remove pass $p_i$}
9:         $C' \leftarrow$ GetCycles$(P, S')$
10:         **if** $C' \leq C_{best}$ **then**
11:             $S \leftarrow S'$ {Accept removal}
12:             $C_{best} \leftarrow C'$
13:             $changed \leftarrow$ **true**
14:         **end if**
15:     **end for**
16: **end while**
17: $S_{min} \leftarrow S$

---

only passes strictly necessary for performance. Application of this strategy to our training corpus reduced the average sequence length from 18.5 to 4.5, a 73.7% reduction.

Figure 3 illustrates the distribution of the reduction ratio across the dataset. We observe that for over 60% of the programs, the redundancy exceeds 70%, confirming that initial search-based sequences are heavily over-provisioned. This substantial compression indicates that performance gains for most programs are driven by a small subset of critical passes. Isolating these core passes is a prerequisite for learning causal efficacy, as it forces the LLM to focus on the high-entropy mappings between program features and specific, impactful transformations. Furthermore, these minimized sequences provide a cleaner ground truth for the Intent-Guided GA, ensuring that the evolutionary search starts from a lean, high-quality prior.

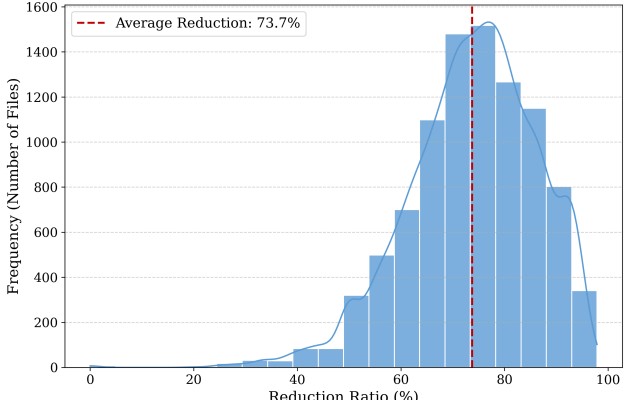

*Figure 3.* The histogram illustrates the percentage of passes removed from original sequences without performance loss.

### 3.1.2. FORENSIC RECONSTRUCTION OF OPTIMIZATION

With the minimal optimal sequences $S_{min}$ established, we construct a forensic analysis framework to capture the evidence explaining the effectiveness of these sequences. We treat optimization as a trajectory of state transitions. For every step $t$ in $S_{min}$, we apply pass $p_t$ to the IR and extract three categories of evidence:

- **Structural Evidence ($\delta_{struct}$):** The textual difference between the pre-pass and post-pass IR, capturing microscopic transformations.

- **Feature Evidence ($\delta_{feat}$):** High-dimensional feature vectors extracted via Autophase(see A) (Haj-Ali et al., 2020). The shift vector $\delta_{feat} = \phi_{t+1} - \phi_t$ quantifies the macroscopic impact of the pass on program statistics.

- **Performance Evidence ($g_t$):** The marginal cycle count improvement after each step. This metric isolates the contribution of individual passes, distinguishing high-impact transformations from setup operations.

Furthermore, we perform a synergy analysis by perturbing the order of adjacent passes. If swapping a pair $(p_i, p_{i+1})$ degrades performance, we label the original order as exhibiting *positive synergy*. This provides the model with negative logic, indicating strict ordering constraints.

### 3.1.3. SIMULATED REASONING FOR DISTILLATION

The collected evidence trace provides a rich knowledge source; however, it cannot be directly used as input for the inference model, as dynamic metrics (e.g., cycles, IR differences) are unavailable for unseen programs. To bridge this gap, we employ a rationale distillation strategy via instruction tuning.

We construct the training dataset by prompting a teacher model (Claude-4.5-Sonnet) to perform a task we term *simulated predictive reasoning*(see B). The objective is to force the model to internalize the causal links between static features and dynamic outcomes using a two-step inference process:

**1. Internal Forensic Analysis:** The teacher model is provided with the full evidence packet, including privileged information such as step-by-step feature deltas ($\Delta\Phi_t$), performance gains ($g_t$), and synergy signals. It analyzes this data to understand the ground truth causality of the optimization trajectory.

**2. Predictive Narrative Generation:** The model generates a rationale $R$ that explains the optimal sequence. Crucially, the prompt enforces a constraint where the narrative must appear to be derived solely from the initial static features $\Phi_{initial}$. The model is prohibited from explicitly mentioning the dynamic evidence files; instead, it must formulate the ground truth feature deltas and performance gains as expert predictions. For example, if the evidence shows a reduction in branch instructions, the model must predict this outcome based on the initial high branch density.

This process generates a dataset of pairs $(\Phi_{initial}, R \oplus S_{opt})$, where $R$ is a chain-of-thought rationale that accurately predicts the structural impact of passes. We fine-tune our target LLM on this dataset using the standard causal language modeling objective:

$$\mathcal{L} = -\sum_t \log P(y_t | \Phi_{initial}, y_{<t}) \qquad (1)$$

where $y$ is the concatenation of the rationale $R$ and the sequence $S_{opt}$. By training on these simulated predictions, the target model learns to act as a simulator: during inference, it utilizes static features to hallucinate correct intermediate states and performance outcomes, using this internalized simulation to guide sequence generation.

### 3.2. Two-Stage Policy Optimization

We employ a two-stage training pipeline using the Qwen2.5-Instruct architecture, integrating SFT with RL to transition from imitation to exploration.

**Stage 1: SFT.** We initialize the policy $\pi_\theta$ on the evidence-driven dataset to establish a causal reasoning prior. This phase achieves two critical objectives: **(1) Causal Alignment:** It conditions the model to generate optimization decisions only after articulating a valid rationale based on static features, enforcing the reasoning bottleneck proposed in Section 3.3. **(2) Format Standardization:** It compels the model to separate internal reasoning from the final decision, ensuring the output is structurally parseable.

**Stage 2: RL via GRPO.** To enable generalization to unseen programs, we further optimize the policy using GRPO (Shao et al.). Unlike methods requiring a separate value network, GRPO normalizes advantages within a group of sampled outputs to guide exploration. The optimization is driven by a composite reward function consisting of two components. **(1) Format Reward ($r_{format}$):** A discrete constraint that validates the output structure. The model is rewarded only if it generates distinct `<think>` and `<answer>` tags and the content within `<answer>` is a syntactically valid JSON array of LLVM pass flags. **(2) Performance Reward ($r_{perf}$):** A continuous signal quantifying optimization quality. For a generated sequence $S_{gen}$ with cycles $C_{gen}$, we define the reward as the relative speedup against the LLVM `-O3` baseline ($C_{O3}$):

$$r_{perf} = \alpha \cdot \frac{C_{O3} - C_{gen}}{C_{O3}} \qquad (2)$$

where $\alpha$ is a scaling coefficient used to normalize the reward magnitude. This formulation directly incentivizes the discovery of sequences that outperform the baseline, aligning the model's exploration with the system's efficiency goals.

### 3.3. Collaborative Inference

While the trained LLM can generate high-quality optimization sequences, the combinatorial nature of the problem makes single-shot prediction brittle. This limitation stems not from weak semantic reasoning, but from the challenge of fine-grained combinatorial execution.

We therefore propose a collaborative inference framework that *explicitly decouples semantic intent from combinatorial execution*. Under this *Strategist–Tactician* paradigm, the LLM serves as the **Strategist**, inferring a high-level distribution over optimization intents, while remaining agnostic to exact pass ordering. The GA then acts as the **Tactician**, operating in the original search space and performing precise combinatorial adjustments guided—rather than dictated—by the inferred intent.

#### 3.3.1. CONSTRUCTION OF PRIOR KNOWLEDGE

Before deployment, we construct a global prior of pass efficacy to inform the search. We perform a large-scale ablation study on the training corpus to quantify the marginal contribution of each pass.

For every program $P$ and its optimal sequence $S_{opt}$ in the training set, we iteratively remove each pass $p_i \in S_{opt}$ to generate a perturbed sequence $S'_{\setminus i}$. The marginal benefit $b(p_i)$ is calculated as the performance degradation caused by its removal:

$$b(p_i) = \mathcal{C}_{nor}(P, S'_{\setminus i}) - \mathcal{C}_{nor}(P, S_{opt}) \quad (3)$$

where $\mathcal{C}_{nor}$ denotes the normalized performance score, computed as $(\text{O3\_cycles} - C)/\text{O3\_cycles}$. We aggregate these values to compute the *Global Expected Benefit* $E[b(p)]$ for every pass in the compiler's vocabulary. We then identify the top-$k$ passes with the highest expected benefits for each functional category (e.g., Loop, Scalar, Vectorization), forming a set of **Star Passes**. This data-driven prior ensures that the search algorithm prioritizes passes with historically high utility.

#### 3.3.2. INTENT-GUIDED EVOLUTIONARY SEARCH

During inference, the LLM predicts an initial sequence $S_0$ from which we derive an intent distribution $\mathcal{I} = \{w_c\}$ over optimization categories. We initialize the GA population with $\{S_0\}$ and employ an intent-guided mutation operator.

Crucially, unlike prior methods that perform *hard pruning* (restricting the search space to a fixed subset), ECCO applies a **soft probabilistic bias**. The mutation probability $P_{mut}(p)$ mixes the LLM's intent with a uniform exploration prior:

$$P_{mut}(p) = \epsilon \cdot \frac{1}{|V|} + (1 - \epsilon) \cdot \frac{w_{\text{cat}(p)} \cdot E[b(p)]}{Z} \quad (4)$$

where $\epsilon$ is the exploration rate and $E[b(p)]$ is the global pass benefit, and $Z = \sum_{p' \in V} w_{\text{cat}(p')} \cdot E[b(p')]$ normalizes the strategic distribution. This formulation ensures **ergodicity**: the search is heavily biased toward the LLM's strategic direction (high $w_c$) for efficiency, yet retains a non-zero probability to select any pass. This allows the GA (Tactician) to theoretically recover from strategic errors made by the LLM, fixing hallucinations through stochastic execution.

## 4. Experimental Setup

**Datasets.** Following GRACE (Pan et al., 2025d), we construct our dataset from the CompilerGym (Cummins et al., 2022) corpus (see C), yielding **9,327** high-quality samples. Evaluation is conducted on seven benchmark suites: `blas`, `cbench`, `chstone` (Hara et al., 2008), `mibench` (Guthaus et al., 2001), `npb` (Bailey et al., 1991), `opencv` (Culjak et al., 2012), and `tensorflow` (Abadi et al., 2016).

**Models and Infrastructure.** We use Qwen2.5-Instruct models at 1.5B, 3B, and 7B scales. All experiments run on Intel Xeon Gold 6430 servers with $4\times$ NVIDIA H100 GPUs. The compiler infrastructure is based on LLVM version 10.0.0.

**Evolutionary Parameters.** We set population size to 50, generations to 20, mutation rate to 1.0, crossover rate to 0.1, top-$k$ stars to 30, maximum sequence length to 120, and exploration rate to 0.2. The mutation rate of 1.0 is used to ensure active sequence-level exploration in every generation and reduce premature stagnation.

**Metric.** Optimization effectiveness is measured by the relative reduction in cycles over LLVM `-O3`, defined as $\mathcal{I}_{O3} = (C_{O3} - C_{opt})/C_{O3}$, where cycles are estimated using `llvm-mca`.

## 5. Experiments

We structure our experimental analysis to answer the following research questions: **RQ1:** How does ECCO compare against state-of-the-art traditional search heuristics and general-purpose LLM prompting? **RQ2:** How does the real-world runtime of ECCO compare to that of state-of-the-art baselines? **RQ3:** How do evidence-driven training, model scaling, and sampling budgets affect optimization efficacy, and what is the quantifiable contribution of the collaborative inference framework? **RQ4:** How can the interpretability of the optimization process be measured?

*Table 1.* Comparison of optimization performance ($\mathcal{I}_{O3}$ in %) across seven benchmark suites. ECCO is compared against Traditional Search Heuristics (top block) and Direct LLM Prompting methods (middle block). The best result in each column is marked in **bold**.

| Method | blas | cbench | chstone | mibench | npb | opencv | tensorflow | Average |
|---|---|---|---|---|---|---|---|---|
| *Traditional Search Heuristics* | | | | | | | | |
| TPE (Bergstra et al., 2011) | 13.45 | 28.60 | 26.07 | 20.70 | 27.85 | 13.40 | 9.07 | 19.88 |
| RIO (Chen et al., 2012) | 16.55 | 30.56 | 27.07 | 22.74 | 31.59 | 15.44 | 9.88 | 21.98 |
| OpenTuner (Ansel et al., 2014) | 15.72 | 31.68 | 27.03 | 22.93 | 32.41 | 15.50 | 9.71 | 22.14 |
| GA (Garciarena & Santana, 2016) | 16.48 | 30.30 | 27.07 | 22.80 | 32.77 | 16.00 | 9.58 | 22.14 |
| PDCAT (Zhu et al., 2025) | 17.19 | 31.75 | 27.84 | 23.44 | 32.78 | 16.19 | 10.33 | 22.79 |
| CompTuner (Zhu et al., 2024) | **17.26** | 31.57 | 27.87 | 23.03 | 31.43 | **18.15** | 10.83 | 22.88 |
| GRACE (Pan et al., 2025d) | 13.72 | 34.08 | 32.95 | 24.69 | 29.31 | 14.98 | 12.25 | 23.14 |
| CFSAT (Pan et al., 2025c) | 16.44 | 31.08 | 28.87 | 24.98 | **34.65** | 17.36 | **12.50** | 23.70 |
| *Direct LLM Prompting(Best-of-32)* | | | | | | | | |
| Qwen3-Coder (Yang et al., 2025) | 8.46 | 27.11 | 27.88 | 20.41 | 15.54 | 9.93 | 4.39 | 16.25 |
| Kimi-K2 (Team et al., 2025) | 10.79 | 24.90 | 28.85 | 20.04 | 18.35 | 10.04 | 3.31 | 16.61 |
| DeepSeek-V3.2 (Liu et al., 2025) | 6.41 | 28.16 | 29.70 | 20.04 | 16.68 | 8.42 | 3.01 | 16.06 |
| GPT5-chat (Jaech et al., 2024) | 10.62 | 26.30 | 27.61 | 19.24 | 18.02 | 10.63 | 3.51 | 16.56 |
| **ECCO** (Qwen2.5-7B, Best-of-32) | 12.99 | **35.19** | **35.50** | **27.12** | 32.97 | 15.58 | 11.72 | **24.44** |

*Table 2.* Runtime reduction (% relative to `-O3`) of CFSAT and ECCO on PolyBench under different tuning budgets.

| Method | 5 min | 10 min | 20 min |
|---|---|---|---|
| CFSAT | 42.87% | 44.89% | 49.48% |
| ECCO | 44.54% | 47.31% | 49.65% |

## 5.1. Overall Optimization Efficacy (RQ1)

**Results.** Table 1 compares performance across seven benchmark suites. ECCO achieves an average cycles reduction of **24.44%** over LLVM `opt -O3`, outperforming CFSAT (23.70%) and GRACE (23.14%), with the largest gains on `cbench`, `chstone`, and `mibench`. In contrast, direct LLM prompting underperforms even with Best-of-32 sampling, with all models clustered around 16.0–16.6% and failing to surpass TPE (19.88%). We additionally compare against other search heuristics. Monte Carlo Tree Search achieves 23.69%, Simulated Annealing achieves 23.07%, and Ant Colony Optimization achieves 24.70% in terms of cycles reduction.

**Analysis.** These findings reveal distinct mechanisms governing optimization efficacy. (1) **Synergy over Blind Search:** ECCO succeeds by addressing the blindness of traditional heuristics. While algorithms like GA and CFSAT excel at local exploitation, they lack the semantic understanding to navigate the global search space efficiently. ECCO utilizes the LLM as a *Strategist* to identify high-potential subspaces based on code semantics, allowing the traditional *Tactician* (GA) to converge faster and more accurately. (2) **The Ceiling of General-Purpose Reasoning:** The suboptimal and uniform performance of the Direct LLM baselines indicates a capability ceiling for general-purpose models in phase-

ordering tasks. Without domain-specific evidence-driven training, even advanced models rely on surface-level pattern matching rather than causal deduction. The fact that GPT-5 performs similarly to Qwen3 suggests that increasing general reasoning capability does not automatically translate to compiler mastery; explicit alignment with IR semantics may be a requisite factor for breaking the performance barrier.

## 5.2. Runtime Comparison (RQ2)

To complement the proxy-based evaluation, we additionally compared ECCO and the best-performing traditional baseline from RQ1 (CFSAT) on real end-to-end execution time using the PolyBench suite. Specifically, we used the 29 programs in PolyBench that can be successfully compiled and executed on our machine. Runtime reduction relative to the `-O3` baseline is reported under different tuning budgets.

Table 2 summarizes the results. Under the 5-minute, 10-minute, and 20-minute budgets, CFSAT achieves runtime reductions of 42.87%, 44.89%, and 49.48%, respectively. In comparison, ECCO achieves 44.54%, 47.31%, and 49.65% under the same budgets.

These results demonstrate that the advantage of ECCO is not limited to MCA-based proxy evaluation. The approach translates to better real execution performance under small-to-medium tuning budgets, and remains slightly better even at the 20-minute budget.

## 5.3. Mechanism Analysis (RQ3)

In this section, we disentangle the contributions of our training strategies and model scaling. Crucially, to evaluate the reasoning capability of the LLM policy itself, we exclude the GA in this entire subsection. All results reported below

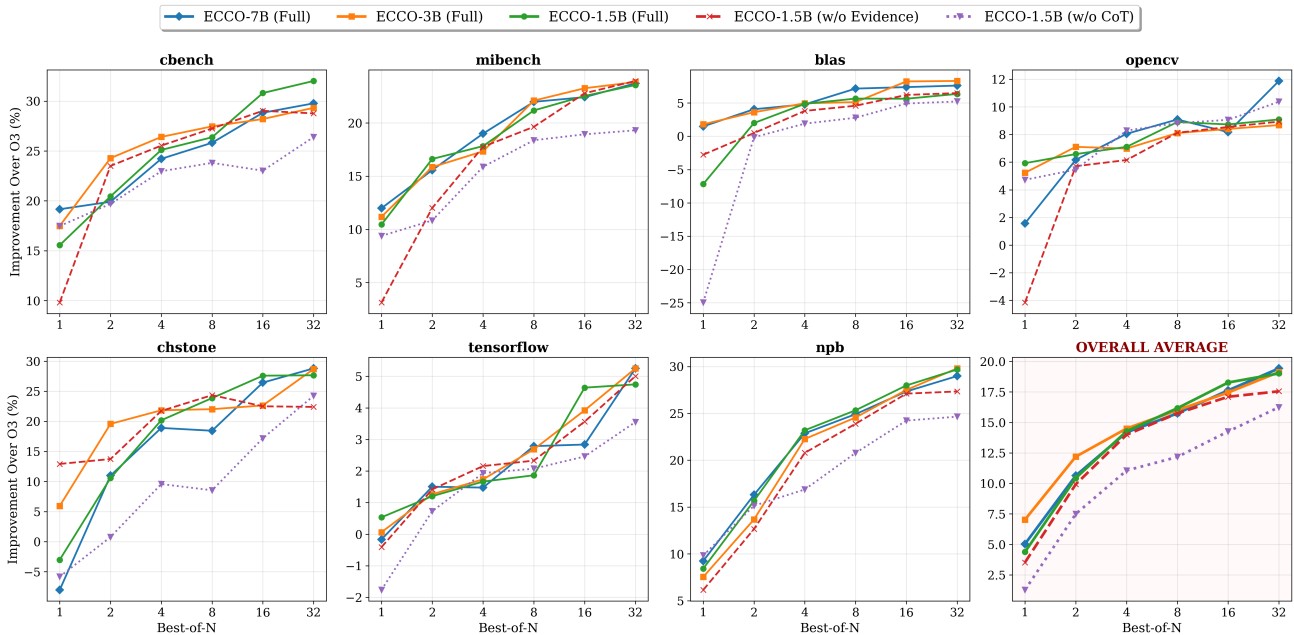

*Figure 4.* **Scaling Analysis across Benchmarks.** We plot the Best-of-$N$ performance ($\mathcal{I}_{O3}$) for different model scales (1.5B, 3B, 7B) and ablated variants. The bottom-right plot shows the aggregated average. Note that the performance of the 1.5B, 3B, and 7B models converges at higher sampling budgets ($N = 32$), while removing Evidence or CoT results in a consistent performance deficit.

*Table 3.* Mechanism Analysis of the LLM policy. We report the improvement over -O3 (%) for each benchmark at $N = 1$ (Greedy) and $N = 32$ (Best-of-N). **Abbreviations:** CB: cbench, MB: mibench, BL: blas, OC: opencv, CS: chstone, TF: tensorflow, NP: npb.

| Configuration | Performance @ $N = 1$ (Greedy) | | | | | | | | Performance @ $N = 32$ (Best-of-N) | | | | | | | |
|---|---|---|---|---|---|---|---|---|---|---|---|---|---|---|---|---|
| | CB | MB | BL | OC | CS | TF | NP | Avg. | CB | MB | BL | OC | CS | TF | NP | Avg. |
| *Ablation on Training Components (1.5B)* | | | | | | | | | | | | | | | | |
| w/o CoT | 17.48 | 9.38 | -24.95 | 4.72 | -5.83 | -1.77 | 9.83 | 1.27 | 26.40 | 19.31 | 5.22 | 10.37 | 24.24 | 3.55 | 24.65 | 16.25 |
| w/o Evidence | 9.81 | 3.12 | -2.77 | -4.15 | 12.92 | -0.41 | 6.15 | 3.52 | 28.76 | 23.94 | 6.49 | 8.93 | 22.39 | 5.01 | 27.34 | 17.55 |
| ECCO-1.5B (w/o GA) | 15.56 | 10.45 | -7.18 | 5.93 | -3.02 | 0.53 | 8.43 | 4.38 | **32.02** | 23.56 | 6.36 | 9.09 | 27.65 | 4.74 | 29.69 | 19.02 |
| *Impact of Model Scaling (LLM Only)* | | | | | | | | | | | | | | | | |
| ECCO-3B (w/o GA) | 17.49 | 11.18 | **1.80** | **5.23** | **5.93** | **0.06** | 7.54 | **7.03** | 29.33 | **23.84** | **8.31** | 8.68 | 28.69 | 5.25 | **29.80** | 19.13 |
| ECCO-7B (w/o GA) | **19.16** | **11.98** | 1.49 | 1.57 | -8.03 | -0.16 | **9.25** | 5.04 | 29.78 | 23.72 | 7.61 | **11.87** | **28.80** | 5.25 | 28.98 | **19.43** |

(Table 3 and Figure 4) represent the performance of the standalone LLM inference. This isolation ensures that the metrics reflect the quality of the learned priors rather than the compensatory effects of stochastic search.

**Results.** Three patterns emerge from the data. First, **Standard ECCO-1.5B (w/o GA)** consistently outperforms ablated variants: removing CoT causes the worst performance (greedy $N = 1$ drops to 1.27%), and removing evidence also reduces results. Second, **ECCO-7B (w/o GA)** reaches the highest peak at $N = 32$ (19.43%), only slightly above 1.5B (19.02%) and 3B (19.13%). Third, all models improve logarithmically with $N$ (Figure 4), with 1.5B, 3B, and 7B converging quickly; in contrast, the gap between Standard ECCO and ablated variants remains across all $N$.

**Analysis.** These results yield three key insights into the mechanism of LLM-based optimization. **(1)** The consistent superiority of the Standard model over the *w/o Evidence* and *w/o CoT* variants confirms that causal reasoning, rather than surface-level pattern matching, is the dominant performance driver. The degradation in the *w/o Evidence* variant underscores the necessity of grounding decisions in verifiable state transitions. Training on forensic signals (e.g., IR structural changes and Autophase deltas) enables the model to anticipate the concrete effects of transformations; without this supervision, it fails to distinguish high-impact passes from redundant ones, producing sequences that are plausible yet semantically ineffective. **(2)** The convergence of the 1.5B, 3B, and 7B curves reveals a capacity–generalization trade-off under a fixed training budget. The regression of the 7B model at $N = 1$ suggests overfitting, likely due to memorization of training noise rather than causal structure. The 3B model emerges as a sweet spot, balancing reasoning capacity and stability. The eventual convergence

*Table 4.* Interpretability audit results. Values show the percentage (%) of ECCO-generated rationales judged as factually consistent with execution evidence by five independent LLM judges. **Avg.** denotes the mean accuracy across benchmarks for each judge.

| Dataset | DeepSeek-V3.2 | Claude-4.5-Sonnet | Qwen3-Coder | GPT5-chat | Kimi-K2 | Consensus Avg. |
|---|---|---|---|---|---|---|
| blas-v0 | 65.52 | 72.41 | 93.10 | 93.10 | 100.00 | 84.83 |
| cbench-v1 | 100.00 | 100.00 | 100.00 | 100.00 | 100.00 | 100.00 |
| chstone-v0 | 100.00 | 100.00 | 100.00 | 100.00 | 100.00 | 100.00 |
| mibench-v1 | 92.31 | 95.00 | 100.00 | 100.00 | 100.00 | 97.46 |
| npb-v0 | 77.50 | 75.00 | 96.43 | 97.37 | 100.00 | 89.26 |
| opencv-v0 | 56.25 | 65.63 | 96.88 | 96.88 | 100.00 | 83.13 |
| tensorflow-v0 | 61.11 | 72.22 | 88.89 | 92.22 | 98.89 | 82.67 |
| **Total Average** | **78.96** | **82.89** | **96.47** | **97.08** | **99.84** | **91.05** |

at $N = 32$ further indicates that increased sampling acts as a de-noising mechanism, allowing over-parameterized models to bypass noisy top-1 predictions and recover valid strategies from the training distribution. **(3)** *Quantifying the Tactical Gap.* Isolating the LLM policy reveals a performance ceiling of $\sim$**19.4%**, even with 7B parameters and extensive sampling. In contrast, the full collaborative framework achieves **24.44%** (RQ1), exposing a $\sim$5% gap. This demonstrates that while the LLM serves as an effective strategist by identifying a high-quality search region, it lacks the combinatorial precision required to reach the global optimum. Removing the GA thus exposes the limits of pure generative modeling and validates the necessity of the proposed Strategist–Tactician collaboration.

### 5.4. Interpretability and Rationale Faithfulness (RQ4)

To answer RQ3, we assess whether the optimization narratives generated by ECCO are factually grounded. We employ an **Evidence-Based LLM-as-a-Judge** framework. Specifically, for every optimization trajectory generated by ECCO-1.5B, we collect the ground-truth evidence using the pipeline described in Section 3.1. We then task five distinct state-of-the-art LLMs to audit the validity of ECCO's CoT, verifying if the predicted effects in the rationale match the physical reality of the compilation execution.

**Results.** Table 4 summarizes the faithfulness accuracy across seven benchmarks as evaluated by different judges. The results indicate a high degree of logical consistency. On logic-intensive benchmarks like cbench and chstone, ECCO achieves nearly **100%** faithfulness across all judges, implying that its reasoning perfectly mirrors the compiler's behavior. Even under the strictest evaluation (DeepSeek-V3.2), the system maintains a minimum average accuracy of **78.96%**. The consensus across all five judges (averaging **91.24%**) confirms that the vast majority of the generated CoT paths are grounded in verifiable causality.

**Analysis.** The high fidelity scores suggest that evidence-driven training effectively bridges the semantic gap between code text and compiler semantics. (1) **Causal Alignment:**

The near-perfect scores on chstone and cbench demonstrate that for well-defined algorithmic kernels, ECCO does not merely guess passes; it accurately predicts their consequences. (2) **Complexity Variance:** We observe a performance dip in opencv and tensorflow under strict judges (DeepSeek/Claude). These benchmarks involve complex vectorization and memory patterns where the Autophase features might be less descriptive, making it harder for the model to articulate a precise rationale that satisfies a rigorous auditor. However, even in these edge cases, the rationale remains plausible, suggesting the reasoning is directionally correct if not perfectly precise.

**Limitations.** We acknowledge that using LLMs to evaluate LLMs introduces potential bias. A judge model might hallucinate or be overly lenient (as seen with Kimi-K2's near-perfect scores). However, the use of a **Multi-Judge Ensemble** mitigates this risk. The fact that even the most critical auditor validates nearly 80% of the samples provides a strong lower bound, confirming that ECCO's interpretability is a structural capability rather than a statistical fluke.

## 6. Discussion

Our experimental findings offer four critical insights into the mechanism of LLMs for systems optimization.

**1. Evidence Acts as a Physical Anchor.** The superior performance of the standard model over the *w/o Evidence* variant indicates that effective CoT data must be grounded in physical reality. Without training on verifiable forensic evidence, the model produces reasoning that is coherent but semantically misaligned with compiler behavior.

**2. Reasoning Structure Outweighs Imperfect Logic.** Notably, the *w/o CoT* variant (pure pattern matching) performed worse than the *w/o Evidence* variant. This implies that the explicit structure of chain-of-thought reasoning acts as a powerful regularizer. Even if the intermediate reasoning lacks ground-truth evidence (as in *w/o Evidence*), the forced decomposition of the problem prevents the model from reverting to fragile surface-level imitation, proving that imperfect reasoning is superior to blind intuition.

**3. Sampling is a Necessity, Not a Luxury.** The scaling analysis reveals that zero-shot ($N = 1$) inference is insufficient for the non-convex optimization space of compilers. However, with Best-of-32 sampling, the pure LLM policy approaches the performance of robust traditional baselines like TPE. This suggests that in the compiler domain, the LLM functions best as a probabilistic generator of diverse candidates. Best-of-$N$ sampling is therefore not merely an enhancement but a fundamental requirement to bypass local minima and exploit the model's learned distribution.

**4. The Last Mile Tactical Gap.** Perhaps the most significant finding is the performance ceiling of the standalone LLM ($\sim 19.4\%$) versus the collaborative system ($24.44\%$). This $\sim 5\%$ differential represents a Tactical Gap. While LLMs excel at *Global Navigation*—identifying the correct strategic vicinity—they lack the combinatorial precision for fine-grained pass ordering. We validate this architectural choice in Appendix E, where we replaced the GA with an iterative LLM refiner. This approach saturated at $\sim 19.36\%$, failing to surpass the simple sampling baseline of the small model. This empirically confirms that traversing the tactical last mile may require rigorous combinatorial search, not just more semantic reasoning.

## 7. Conclusion

In this paper, we presented ECCO, a framework that bridges semantic reasoning and combinatorial search for compiler auto-tuning. By shifting from black-box search to evidence-driven causal reasoning, ECCO overcomes key limitations of traditional heuristics and existing LLM-based approaches. Our approach constructs a high-quality Chain-of-Thought dataset via reverse engineering, enabling the model to capture causal links between code features and performance. Moreover, the proposed *Strategist–Tactician* framework combines the global planning ability of LLMs with the local precision of evolutionary search. Experimental results show that ECCO achieves an average 24.44% reduction in cycles compared to LLVM `opt -O3`, consistently outperforming direct LLM prompting and strong traditional baselines. Importantly, ECCO produces verifiable optimization narratives, advancing transparency and interpretability in compilation.

## Impact Statement

This paper presents work whose primary goal is to advance the field of machine learning by improving methodological understanding and system-level performance. The techniques introduced in this work are intended for general research and engineering use, and we do not foresee direct negative societal impacts arising from their application. As with most advances in machine learning, the methods proposed here could be incorporated into downstream systems whose societal effects depend on the specific application context. We emphasize that responsible deployment, including appropriate evaluation, monitoring, and human oversight, remains essential when applying machine learning techniques in real-world settings.

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

# Appendix

## A. AutoPhase Feature Set

Our framework utilizes the 56 statistical features from AutoPhase (Haj-Ali et al., 2020) to represent programs compactly for the LLM. These features capture various aspects of the program's static structure and instruction mix.

| Index | Feature Description | Index | Feature Description |
|---|---|---|---|
| 0 | BBs: total phi args > 5 | 28 | Number of And insts |
| 1 | BBs: total phi args in [1,5] | 29 | BBs: instruction count in [15,500] |
| 2 | BBs: count with 1 predecessor | 30 | BBs: instruction count < 15 |
| 3 | BBs: count with 1 predecessor and 1 successor | 31 | Number of BitCast insts |
| 4 | BBs: count with 1 predecessor and 2 successors | 32 | Number of Br insts |
| 5 | BBs: count with 1 successor | 33 | Number of Call insts |
| 6 | BBs: count with 2 predecessors | 34 | Number of GetElementPtr insts |
| 7 | BBs: count with 2 predecessors and 1 successor | 35 | Number of ICmp insts |
| 8 | BBs: count with 2 predecessors and successors | 36 | Number of LShr insts |
| 9 | BBs: count with 2 successors | 37 | Number of Load insts |
| 10 | BBs: count with >2 predecessors | 38 | Number of Mul insts |
| 11 | BBs: Phi node count in range (0,3] per BB | 39 | Number of Or insts |
| 12 | BBs: count with more than 3 Phi nodes | 40 | Number of PHI insts |
| 13 | BBs: count with no Phi nodes | 41 | Number of Ret insts |
| 14 | Number of Phi-nodes at beginning of BB | 42 | Number of SExt insts |
| 15 | Number of branches | 43 | Number of Select insts |
| 16 | Number of calls that return an int | 44 | Number of Shl insts |
| 17 | Number of critical edges | 45 | Number of Store insts |
| 18 | Number of edges | 46 | Number of Sub insts |
| 19 | Occurrences of 32-bit integer constants | 47 | Number of Trunc insts |
| 20 | Occurrences of 64-bit integer constants | 48 | Number of Xor insts |
| 21 | Occurrences of constant 0 | 49 | Number of ZExt insts |
| 22 | Occurrences of constant 1 | 50 | Number of basic blocks |
| 23 | Number of unconditional branches | 51 | Number of instructions (all types) |
| 24 | Binary operations with a constant operand | 52 | Number of memory instructions |
| 25 | Number of AShr insts | 53 | Number of non-external functions |
| 26 | Number of Add insts | 54 | Total arguments to Phi nodes |
| 27 | Number of Alloca insts | 55 | Number of Unary operations |

## B. Prompt Templates

### B.1. Teacher Model Prompt for Data Construction

We utilized Claude-4.5-Sonnet as the teacher model to generate the evidence-driven Chain-of-Thought dataset. The prompt is designed to force the model to internalize the causal link between static features and dynamic outcomes. Table 6 presents the condensed system instruction.

---

**System Role:**
You are a world-class compiler expert with the unique ability to both perform deep, evidence-based analysis and then articulate your findings as concise, high-level expert intuition. Your task is to reverse-engineer the expert thought process behind an optimal pass sequence.

---

**Task Instructions:**
**STEP 1: INTERNAL ANALYSIS (Your Private Scratchpad)**
First, silently and internally, perform a deep analysis of the following complete evidence file. Your goal is to fully understand the *true* causal chain of the optimization. Key evidence fields include:

- `autophase_features`: A dictionary of 56 static features of the original program's LLVM IR.

- `optimal_sequence`: The exact sequence of compiler passes that achieved the best performance.

- `pass_profile`: A list detailing the *immediate* effect of each pass (including `ir_diff` and `autophase_delta`).

- `synergy_analysis`: Interactions between adjacent passes (e.g., Positive Synergy).

[...Full Evidence JSON String...]

**STEP 2: FINAL OUTPUT**
Generate the final output as a single JSON object. **CRITICALLY, your narrative MUST follow the 'feigned reasoning' principle:**

**FEIGNED REASONING PRINCIPLE:**
Your reasoning must *appear* to be a series of intuitive deductions and predictions based *only* on the **initial `autophase_features`**. You are strictly forbidden from directly mentioning low-level evidence like `ir_diff` or `synergy_analysis`. However, you MUST incorporate your internal findings from `autophase_delta` and `performance` by presenting them as your own **expert predictions**.

**NARRATIVE STRUCTURE REQUIREMENTS:**
1. **Initial Diagnosis**: Start with a diagnosis based on the initial features.
2. **Step-by-Step Rationale**: For each pass:

- State the Decision.

- Justify the Decision based on program state.

- **Predict the State Change:** You must predict the key changes to the Autophase features. The values **must exactly match** the `autophase_delta` from the evidence file. Example: "My prediction is that this pass will reduce `BranchCount` by 10."

3. **Final Performance Prediction**: Predict the overall speedup (must match `speedup_over_o3`).
3. **Final Sequence Proposal**: The final JSON array of flags.

---

*Table 6.* The system prompt used to instruct the teacher model (Claude-4.5-Sonnet) for rationale distillation. The prompt enforces a "Feigned Reasoning" constraint, compelling the model to translate privileged dynamic evidence into predictable causal logic based on static features.

## B.2. Interpretability Verification Prompt (LLM-as-a-Judge)

To evaluate the faithfulness of the generated rationales (RQ3), we employ a strong general-purpose LLM (e.g., GPT-4o, DeepSeek-V3) as an auditor. The judge is provided with the *Ground Truth* (actual execution traces, IR diffs, and feature changes) and the *Claim* (ECCO's generated thought process). Table 7 presents the exact prompt used for this verification.

| |
|---|
| **Task Instruction:** |
| Please judge the reasonableness of the thinking analysis (`think_content`) based on the following actual optimization data: |
| **Section 1: Actual Optimization Data (Ground Truth)** |
| **- Final Pass Sequence Used:** `['-pass1', '-pass2', ...]` |
| **- Performance Gain over -O3:** `0.xx` |
| **- Initial Autophase Features (before any optimization pass):** |
| ```json``` |
| `{...Full Dictionary of 56 Features...}` |
| ``` ``` |
| **- Full Evidence Trace (Feature Changes, Perf Gain, IR Diff after each pass):** |
| ```json``` |
| `{...The forensic evidence collected via Section 3.1...}` |
| ``` ``` |
| **- Synergy Analysis between Optimization Passes:** |
| ```json``` |
| `{...Positive/Negative Synergy Pairs...}` |
| ``` ``` |
| **Section 2: Thinking Analysis to be Judged** |
| ```text``` |
| `[Insert the <think> content generated by ECCO-1.5B]` |
| ``` ``` |
| **Output Requirements:** |
| 1. First line: Only 'Correct' or 'Incorrect' |
| 2. Second line onwards: Detailed judgment reason (explain your decision) |
| 3. DO NOT analyze numerical predictions or implementation minutiae |
| 4. DO NOT provide extensive rationale |

*Table 7.* The verification prompt used for the automated interpretability audit. The Judge Model compares the causal claims in the generated rationale against the forensic evidence. Note that the judge is instructed to focus on *logical consistency* and *causal validity* rather than scrutinizing minor numerical precision, ensuring a robust evaluation of the reasoning flow.

## B.3. Student Model Prompt for Inference

Table 8 presents the prompt used for the Student Model (Qwen2.5-Instruct). Unlike the teacher model, the student model does not receive the full evidence trace or source code. Instead, it must rely solely on the static Autophase features to reason about the program's characteristics and generate an optimal pass sequence. This prompt structure is used during both the SFT/RL training phases and the final collaborative inference.

| |
|---|
| **System Role:** 
 You are a world-class compiler optimization expert. Your task is to find the optimal pass sequence for a given LLVM IR program, aiming to **maximize its runtime performance (minimize execution cycles)**. |
| **Input Context:** 
 The program is represented by its static Autophase features. Analyze these features to deduce the program's characteristics and performance bottlenecks. 

 **Program Autophase Features:** 
 ```json 
 [...Serialized Autophase Feature Dictionary...] 
 ``` |
| **Constraints & Vocabulary:** 
 Based on your analysis, provide your final recommended pass sequence. 
 **You MUST select passes only from the following list:** 
 ```text 
 -add-discriminators, -adce, -aggressive-instcombine, ..., -loop-unroll, -loop-vectorize, ..., -tailcallelim, -mergereturn 
 *(Note: The full list contains 100+ valid LLVM pass flags provided in the actual prompt)* 
 ``` |
| **Formatting Instructions:** 
 **Your thought process should be enclosed in `<think>` tags, and your final answer (the pass sequence list) must be enclosed in `<answer>` tags.** 
 Do not invent or use any pass not in the list above. |

*Table 8.* The inference prompt for the Student Model. The model is constrained to a fixed vocabulary of optimization passes and is required to output a Chain-of-Thought rationale (`<think>`) prior to the final sequence (`<answer>`), enforcing the "Strategist" behavior.

## C. LLVM IR Dataset Preparation Details

LLVM IR programs were aggregated from multiple public datasets within CompilerGym. The resulting dataset was partitioned into training and evaluation sets. The training set comprises programs sampled from six **uncurated** datasets, representing broadly collected corpora. Specifically, the raw training corpus contains 19,603 LLVM IR programs. After applying the filtering process, we obtain 9,327 final effective training instances. This reduction mainly comes from two steps: (1) after Algorithm 1, we discard instances with OverO3 $\leq 0$, because ECCO focuses on high-quality optimization evidence; and (2) we remove instances for which the evidence-construction procedure in Sec. 3.1.2 does not complete successfully under resource or context constraints. These **curated** benchmarks are specifically utilized for out-of-distribution assessment and serve as standard collections for rigorous evaluation. Program counts per dataset and split are summarized in Table 9.

*Table 9.* Datasets Summary.

| **Uncurated Datasets** | | | |
|---|---|---|---|
| **Type** | **Dataset** | **Train** | **Test** |
| Uncurated | blas-v0 | 133 | 29 |
| | github-v0 | 7,000 | 0 |
| | linux-v0 | 4,906 | 0 |
| | opencv-v0 | 149 | 32 |
| | poj104-v1 | 7,000 | 0 |
| | tensorflow-v0 | 415 | 90 |

| **Curated Datasets** | | | |
|---|---|---|---|
| **Type** | **Dataset** | **Train** | **Test** |
| Curated | cbench-v1 | 0 | 11 |
| | mibench-v1 | 0 | 40 |
| | chstone-v0 | 0 | 12 |
| | npb-v0 | 0 | 121 |

| **Total** | – | | | | | | **19,603** | **335** |
|---|---|---|---|

## D. Qualitative Case Study: Full Inference Trace

Table 10 presents the complete inference log for the program `blas-v0_70.ll`. The trace demonstrates the model's ability to perform reasoning: it diagnoses bottlenecks (e.g., memory traffic), selects passes (e.g., `-newgvn`, `-sroa`), and explicitly predicts the resulting feature deltas (e.g., `"NumLoadInst":  -10`).

| ECCO-1.5B Optimization Log | |
|---|---|
| **Target File** | `blas-v0_70.ll` |
| **Performance** | **Improvement over -O3:** +27.34% ($\mathcal{I}_{O3} = 0.273$) |
| **Part 1: Initial Diagnosis (Static Analysis)** | |
| **Observation** | • **Control Flow:** Moderate complexity (`TotalBlocks:  33`, `TotalInsts:  197`). High usage of PHI nodes (`NumPHIInst:  16`) indicates significant SSA structures.
• **Memory Bottleneck:** `TotalMemInst:  42` (approx. 21% of all instructions). High load/store ratio indicates redundancy.
• **Vectorization Hints:** `NumICmpInst:  19` and `NumGetElementPtrInst:  14` suggest nested loops with array index computations. |
| **Part 2: Step-by-Step Optimization Strategy** | |
| **Step 1** | **Pass:** `-newgvn` (Global Value Numbering)
*Rationale:* Foundational optimization to identify redundant loads given the high memory traffic.
*Predicted Feature Delta:* `{"NumLoadInst":  -10, "TotalInsts":  -17, "NumGetElementPtrInst":  -5}` |
| **Step 2** | **Pass:** `-slsr` (Scalar Replacement of Aggregates)
*Rationale:* After GVN simplifies expression trees, SLSR performs strength reduction.
*Predicted Feature Delta:* `{"NumPHIInst":  +6, "TotalInsts":  -11}` (Phi increase reflects specialized forms). |
| **Step 3** | **Pass:** `-licm` (Loop Invariant Code Motion)
*Rationale:* Moves invariant computations out of loops to reduce overhead.
*Predicted Feature Delta:* `{}` (Structural improvement not reflected in static instruction counts). |
| **Step 4** | **Pass:** `-sroa` (Scalar Replacement of Aggregates)
*Rationale:* Crucial for promoting stack allocations to registers after previous cleanups.
*Predicted Feature Delta:* `{"NumAllocaInst":  -1, "NumStoreInst":  -10, "NumLoadInst":  -10, "TotalMemInst":  -21}` |
| **Step 5** | **Pass:** `-tailcallelim`
*Rationale:* Optimizes calling conventions, though no immediate tail pattern observed.
*Predicted Feature Delta:* `{}` |
| **Step 6** | **Pass:** `-loop-rotate`
*Rationale:* Optimizes control flow structure for better branch prediction.
*Predicted Feature Delta:* `{"BranchCount":  +2, "TotalBlocks":  +2, "TotalInsts":  +2}` |
| **Part 3: Ordering Logic** | |
| **Logic** | 1. **Foundation:** `-newgvn` must run first to expose redundancies.
2. **Chain Reaction:** `-slsr` and `-licm` rely on the simplified expressions from Step 1.
3. **Memory Promotion:** `-sroa` is placed mid-stream to handle the now-clean representation.
4. **Final Polish:** `-loop-rotate` concludes to restructure control flow. |
| **Part 4: Final Output** | |
| **Sequence** | `["-newgvn", "-slsr", "-licm", "-sroa", "-tailcallelim", "-loop-rotate"]` |

*Table 10.* Detailed inference trace for `blas-v0_70.ll`. The model translates static Autophase features into a precise, causal optimization trajectory, explicitly predicting the impact of each pass on the program's feature vector.

# E. LLM-as-a-Refiner: Expert Iteration Analysis

In this supplementary experiment, we evaluate whether powerful general-purpose LLMs can act as refiners to improve the sequences generated by ECCO-1.5B. We benchmark this approach against two baselines: simple extensive sampling of the specialized small model.

**Experimental Setup and Results.** The refinement pipeline operates as a two-stage feedback loop. First, ECCO-1.5B generates $N$ candidate sequences ($N \in \{1, \ldots, 16\}$), which are executed to record performance and Autophase features. Second, a teacher LLM analyzes this trial history to synthesize a refined best sequence. We evaluated five models on the five common benchmark suites.

*Table 11.* Performance Trend of Refiner LLMs. Values represent **Average OverO3 (%)** across 5 datasets. Without sufficient history ($N \leq 2$), Refiners fail to outperform simple selection.

| Refiner Model | History $N = 1$ | History $N = 2$ | History $N = 4$ | History $N = 8$ | History $N = 16$ |
|---|---|---|---|---|---|
| Qwen3-Coder | 8.56 | 10.45 | 14.82 | **17.95** | **19.36** |
| DeepSeek-V3.2 | 11.74 | 12.44 | 15.69 | 17.68 | 18.60 |
| Claude-4.5-Sonnet | 12.50 | 11.99 | **16.96** | 17.73 | 18.91 |
| GPT-5 | **14.77** | **16.13** | 16.13 | 17.83 | 17.99 |
| Kimi-K2 | 12.85 | 14.30 | 15.65 | 16.40 | 19.19 |

*Table 12.* Performance Comparison ($N = 16$). Increasing the sample size of the small model (Best-of-32) outperforms using a large model to refine the history, while the GA framework remains superior to both.

| Method | blas | cbench | chstone | mibench | opencv | Average |
|---|---|---|---|---|---|---|
| *Large Model as Refiner (Input: History of 16 trials)* | | | | | | |
| GPT-5 | 5.71 | 30.13 | 23.77 | 22.36 | 7.99 | 17.99 |
| DeepSeek-V3.2 | 4.90 | 28.78 | 27.09 | 23.08 | **9.15** | 18.60 |
| Claude-4.5-Sonnet | 6.02 | 30.69 | 27.08 | 22.62 | 8.14 | 18.91 |
| Kimi-K2 | 6.77 | 31.29 | 26.76 | 22.80 | 8.31 | 19.19 |
| Qwen3-Coder | **7.70** | 29.46 | **28.82** | 22.04 | 8.77 | 19.36 |
| *Standalone Small Model (Simple Best-of-N Sampling)* | | | | | | |
| ECCO-1.5B (Best-of-16) | 5.63 | 30.81 | 27.59 | 22.53 | 8.73 | 19.06 |
| **ECCO-1.5B (Best-of-32)** | 6.36 | **32.02** | 27.65 | **23.56** | 9.09 | **19.74** |

**Analysis of Refinement Efficacy and Model Behavior.** First, regarding the **magnitude of refinement**, we observe a phenomenon of *diminishing marginal gains* relative to the sampling budget. At the cold start phase ($N = 1$), the external refiner provides massive value: the standalone ECCO-1.5B achieves only 4.38%, whereas the GPT-5 augmented to 14.77%, and even the weaker Qwen3-Coder boosts performance to 8.56%. This indicates that with minimal exploration, the reasoning injection from a large model significantly corrects the suboptimal trajectories of the small model. However, as $N$ increases, this advantage evaporates. At $N = 16$, the best refiners (Qwen3-Coder: 19.36%, Kimi-K2: 19.19%) are statistically indistinguishable from the standalone best-of-16 result (19.06%). This implies that as the sampling budget expands, the small model naturally covers the high-quality solution space, rendering the external Refiner redundant—it effectively devolves from a creator of new solutions to merely a selector of existing ones.

Second, regarding **model-specific characteristics**, we observe distinct behaviors in In-Context Learning (ICL). GPT-5 acts as a *Strong Reasoner*: it starts with a high baseline at $N = 1$ (14.77%) due to superior internal knowledge, but gains relatively little from additional history ($N = 16 \rightarrow 17.99\%$). In contrast, Qwen3-Coder behaves as a *Strong Pattern Matcher*: it starts weak (8.56%), but exhibits the steepest learning curve, eventually surpassing GPT-5 to reach 19.36% at $N = 16$. This suggests that while GPT-5 relies on intrinsic semantic understanding, specialized coding models like Qwen3 are highly sensitive to the optimization gradient provided in the history context, allowing them to extrapolate better strategies when sufficient data points are available.

