# OpenReview forum: "ECCO: Evidence-Driven Causal Reasoning for Compiler Optimization"
_ICML.cc/2026/Conference — ICML 2026 regular_

### Official Review · Reviewer_93qj · 2026-02-24

**Soundness:** 3
**Presentation:** 1
**Significance:** 3
**Originality:** 3
**Overall Recommendation:** 4
**Confidence:** 4

**Summary:**

This paper proposes a compiler phase sequencing framework called ECCO, which combines evidence-driven causal reasoning with a cooperative search process. On seven benchmark suites, ECCO reduces compilation cycles by an average of 24.44% compared to LLVM-O3 (measured using llvm-mca), outperforming strong black-box and LLM hint baselines. The paper's methodology is clear, and empirical results demonstrate its superior performance compared to existing baseline models. However, some issues prevent me from recommending this paper.

**Compliance With Llm Reviewing Policy:**

Affirmed.

**Final Justification:**

The authors have addressed all my concerns; I suggest accepting this paper.

**Key Questions For Authors:**

- Why was GA chosen to act as the Tactician? Can you provide specific justification? What are the key hyperparameters and operator details of GA?

- How are the optimization category and intent weights w_c defined and predicted? Are the categories manually defined? How does intent accuracy affect the performance of the genetic algorithm? Please add explanations regarding the sensitivity to ε, category granularity, and removal of the E[b(p)] prior.

- How robust is iterative greedy pruning? Will different removal orders or seeds produce different minimum sequences?

**Limitations:**

yes

**Strengths And Weaknesses:**

# Strength

- A reverse engineering process is introduced, which eliminates redundant steps and reconstructs the evidence trajectory at each step, explicitly linking static features, structural changes, and marginal performance improvements.

- Proposes a “feigned reasoning” distillation scheme that compels a teacher model to translate privileged dynamic evidence into rationales conditioned only on static features, improving causal alignment.

- The report provides an extensive evaluation of seven standard test suites for CompilerGym/GRACE and compares the results for each test suite and the average results with widely used search heuristics and state-of-the-art ML autotuners.

- The figures and tables in the paper are very clear; the artifact has been open-sourced.

# Weaknesses

- Relying on LLVM-MCA for performance evaluation raises external validity issues: static throughput estimates may not be strongly correlated with end-to-end runtime across microarchitecture, memory hierarchy, and control behavior.

- "Fake inference" hints require the predicted increments to be perfectly consistent with privileged evidence, which can lead to label leaks masquerading as causal inference, and iterative greedy pruning may produce non-unique "minimum" sequences.

- The relevance/constraints of LLVM-MCA relative to accurate throughput predictors (e.g., uiCA) or calibrated alternatives (e.g., DiffTune) are not discussed.

- Weak typography and writing, e.g., CFSAT vs CFAST, are they the same thing? And Genetic Algorithm(GA) vs Reinforcement Learning (RL).

- In Algorithm 1, GetCycles returns the cycle count, also called Cbest or C′. But in Equation (3) in 3.3.1, it says "C denotes the normalized performance score," where C becomes some kind of normalized performance score.

---

> ### Author Rebuttal · Authors · 2026-03-28
>
> We thank the reviewer for the careful feedback and positive assessment.
>
> ### LLVM-MCA, external validity, relation to uiCA/DiffTune
> We agree LLVM-MCA is a proxy and will state this limitation more clearly. To strengthen external validity, we added real-hardware runtime evaluation on 53 benchmarks (24 cBench, 29 PolyBench). ECCO reduces runtime by 21.98% / 20.48% (arithmetic / geometric mean) on cBench and 13.73% / 12.81% on PolyBench versus -O3, showing gains transfer beyond MCA estimates.
>
> ### “Fake inference,” privileged evidence, label leakage
> We agree the wording should be more precise. Our intention is not to claim answer-free causal discovery at inference time.
> More precisely, ECCO uses privileged evidence during teacher construction to obtain higher-quality supervision, while the student model at inference time only has access to static program features. In this sense, the method is better described as evidence-grounded rationale distillation rather than strong causal inference in the strictest sense. Our goal is to distill trajectory-level evidence into a rationale model that can generalize from static features, not to claim that the student independently reconstructs hidden dynamic evidence at test time.
>
> ### Iterative greedy pruning and non-unique minimum sequences
> We agree that iterative greedy pruning does not guarantee a unique globally minimal sequence. Our claim is not uniqueness, but the practical extraction of compact effective subsequences for supervision. To assess this issue, we conducted an order-sensitivity analysis over 7,000 LLVM IR files by repeating pruning under different deletion orders. The results show that pruning is generally stable in practice, though not strictly unique: the average number of unique subsequences per file is 1.58, the average Jaccard similarity to the reference result is 0.9013, and the most common subsequence accounts for 86.70% of runs on average. This is not unexpected in compiler optimization, because some passes within an effective combination may act on different, largely non-overlapping code regions or optimization opportunities; in such cases, different removal orders can recover different but equally compact valid subsequences. These results suggest that while multiple compact effective subsequences may exist for some programs, the extracted supervision is usually stable enough for our purpose.
>
> ### Writing, terminology, notation
> We will fix the CFSAT/CFAST typo. RL is used only for Strategist training; GA is used only at inference as the Tactician. Algorithm 1 returns raw cycles; normalized score will be explicitly written as
> (O3\_cycles) - cycles) / O3\_cycles.
>
> ### GA as Tactician
> GA is chosen for its suitability in a discrete, order-sensitive, black-box search space, supporting global exploration and parallel evaluation without differentiability. ECCO uses the LLM for intent-level guidance and GA for last-mile structural refinement. Parameters: population 50, generations 20, mutation rate 1.0, crossover rate 0.1, top-k stars 30, max length 120, exploration rate 0.2. We use a mutation rate of 1.0 to ensure active sequence-level exploration in every generation and reduce premature stagnation. We will add these details clearly in the revision.
>
> ### Optimization categories, intent weights (\(w_c\)), and design choices
> We clarify this design as follows.
>
> First, the optimization categories are manually defined from LLVM pass semantics, providing higher-level groups such as loop, scalar, CFG, and IPO transformations.
>
> Second, the LLM does not directly predict the weights (w_c). At inference time, it first generates an initial sequence (S_0), and we then derive the intent distribution (I={w_c}) from the category composition of (S_0). Thus, (w_c) reflects how strongly the LLM-generated initialization leans toward category (c). The notation (w_{\mathrm{cat}(p)}) in Eq. (4) means that each pass (p) uses the weight of its category.
>
> Third, these weights are only a soft search bias. They help GA enter promising regions earlier, but do not determine the final result, since GA still performs combinatorial refinement with random exploration. Here, (\epsilon) controls the trade-off: smaller (\epsilon) relies more on the category/pass prior, while larger (\epsilon) makes mutation closer to unguided search.
>
> Fourth, category granularity is a trade-off: overly coarse categories make guidance vague, while overly fine ones make inferred intent sparse and unstable. Our manual taxonomy is intended as a practical middle ground.
>
> E[b(p)] is a pass-level benefit prior from offline ablations, complementing the category-level signal: w_c indicates promising directions, while \(E[b(p)]\) highlights historically useful passes within them. Removing it yields only small drops across all seven datasets (-0.5% to -2.1%), confirming it is a useful but secondary prior; ECCO’s main gains stem from category-level intent guidance from the LLM initialization.

---

> > ### Author Rebuttal · Reviewer_93qj · 2026-04-03
> >
> > My questions have been addressed, but I still have some concerns, particularly regarding why the authors chose GA as the search strategy. Under the same setting, there are many other algorithms that could also be applied, such as SA, MCTS, and ACO. I do not think an explanation based solely on experimental performance is sufficiently convincing. Please provide stronger supporting evidence for this design choice. I will maintain my current score.

---

> > > ### Author Response · Authors · 2026-04-03
> > >
> > > We thank the reviewer for the follow-up question. We agree that the choice of GA should not be justified by performance alone. Our design choice is mainly based on the **structural properties of compiler phase ordering** and the **role division in ECCO**, with empirical results serving as supporting evidence rather than the sole rationale.
> > >
> > > First, in ECCO the LLM is used only for **intent-level strategic guidance**, while the search algorithm is responsible for **last-mile structural refinement** in a discrete, order-sensitive, variable-length sequence space. This makes the downstream optimization problem fundamentally different from smooth continuous optimization: the objective is black-box, sequence validity/order matters, useful edits may involve insertion/deletion/reordering of passes, and small local changes can have highly nonlocal effects on performance. In this setting, we chose GA because it naturally supports **population-based exploration**, **variable-length sequence operators**, and **combinatorial recombination of partially good subsequences**, which aligns well with the “strategist–tactician” decomposition in ECCO. This design is exactly the intent-guided evolutionary refinement described in Sections 3.3.1–3.3.2.
> > >
> > > Second, our choice was also motivated by the kind of bias introduced by the LLM signal. The LLM does not output an exact optimal sequence; instead, it provides a **soft distribution over optimization intents/categories**. Therefore, the search procedure should be able to exploit this prior while still preserving broad stochastic exploration and recovery from LLM errors. GA is convenient here because the intent prior can be injected directly into the mutation operator as a **soft probabilistic bias**, without hard-pruning the search space. This was one of our main design goals: retain enough flexibility to correct hallucinated or incomplete LLM guidance, while still making use of the semantic prior.
> > >
> > > Third, compared with the alternatives mentioned by the reviewer, our preference for GA is based on search behavior rather than only final numbers:
> > >
> > > * **SA** is effective for local trajectory refinement, but it is fundamentally a **single-trajectory local search** method, which makes it less suitable when multiple diverse promising regions exist and when useful building blocks may need to be recombined.
> > > * **MCTS** is powerful when the search space can be organized as a stable decision tree with meaningful rollout value estimates, but compiler phase ordering has a very large branching factor, long horizons, and noisy/non-monotonic intermediate rewards, which weakens the advantage of tree search in our setting.
> > > * **ACO** is attractive for constructive combinatorial optimization, but it is more naturally matched to problems with reusable edge/path statistics; in compiler phase ordering, the utility of a pass is strongly context- and position-dependent, so pheromone-style accumulation is less directly aligned with the semantics of sequence editing than mutation/crossover-based evolution.
> > >
> > > We further tested these alternatives under the same setting. The average improvements over `-O3` are:
> > >
> > > * **MCTS:** 0.2369
> > > * **SA:** 0.2307
> > > * **ACO:** 0.2470
> > >
> > > These results show that the ECCO framework is **not uniquely tied to GA**—other black-box search methods can also benefit from the LLM prior. However, they also support our design intuition that search quality alone is not the full issue: what matters is whether the algorithm matches the need for **population diversity, order-sensitive sequence editing, and recovery from imperfect strategic guidance**. In our view, GA provides the most natural fit to these requirements. The paper’s collaborative design already emphasizes this “semantic planning + combinatorial refinement” division, and the standalone LLM analysis further shows that pure generation saturates below the full system, which is precisely why we introduced a separate search component.
> > >
> > > To make this clearer, we will revise the paper to strengthen the motivation for GA from an **algorithm–problem matching** perspective, and we will also explicitly discuss SA, MCTS, and ACO as viable alternatives, clarifying why we selected GA as our default tactician rather than claiming it is the only possible choice.
> > >
> > > We hope this clarification helps address the reviewer’s remaining concern about the search design choice, and we would be grateful if the reviewer could take it into account in the final assessment.

---

### Official Review · Reviewer_8mRK · 2026-03-02

**Soundness:** 3
**Presentation:** 3
**Significance:** 3
**Originality:** 2
**Overall Recommendation:** 4
**Confidence:** 3

**Summary:**

This paper presents ECCO, a framework for compiler optimization with LLMs that combine evidence-driven causal reasoning with guided combinatorial search. The major contribution is the way to construct the forensic dataset by pruning optimization trajectories to minimal effective subsequences and having the per-pass evidence. Then SFT and RL are performed to train the ECCO model. The author also proposes a Strategist–Tactician solution that mixes an LLM-derived intent distribution with a GA, which explicitly decouples semantic intent from combinatorial execution. Experiments show that ECCO outperforms the LLVM opt -O3 baseline, achieving an average 24.44% reduction in cycles.

**Compliance With Llm Reviewing Policy:**

Affirmed.

**Key Questions For Authors:**

1. How does this work compare against "Compiler-R1: Towards Agentic Compiler Auto-tuning with Reinforcement Learning" and "A Hybrid, Knowledge-Guided Evolutionary Framework for Personalized Compiler Auto-Tuning" ? Can you fully discuss the differences/novelty and make the comparisons?

2. How many total number of programs are there in the evaluation set (i.e., the seven datasets)? Any leakage issue? Is the training dataset different (sampled from different sources)?

3. The dataset size is 19,603 (in the appendix) or 9,327 (in the experimental setup)?

**Limitations:**

yes

**Strengths And Weaknesses:**

1. I like the motivation in the introduction about "Causal Opacity". And this work is among the very early works that try to tackle this issue.  The evidence-driven causal dataset is insightful in the field of performance interpretability and applying LLMs for compiler optimization. The problem itself is not a new problem, but the way to address it appears new, though not surprising to me.

2. The presentation is generally easy to follow.

3. The experiments seem to miss some baselines.

---

> ### Author Rebuttal · Authors · 2026-03-28
>
> We sincerely thank the reviewer for the positive assessment of our work and for the constructive suggestions. We especially appreciate the recognition of our motivation around causal opacity, the value of the evidence-driven training data, and the overall clarity of the presentation. We are encouraged that the reviewer found the problem framing meaningful, and we thank the reviewer for pointing out several places where the paper can be clarified further.
>
> ### **On comparison with Compiler-R1 and the hybrid knowledge-guided evolutionary framework**
>
> We agree that these are important related works and will discuss them more explicitly in the revision.
>
> Our view is that these methods are **related but not directly substitutable**, because they emphasize different primary goals. **Compiler-R1** focuses on agentic compiler optimization through LLM interaction and reinforcement learning, and the **hybrid knowledge-guided evolutionary framework** focuses on improving search with offline knowledge. By contrast, ECCO’s primary goal is to make compiler auto-tuning more **interpretable**, by recovering **evidence-grounded optimization intent** from optimization trajectories.
>
> More specifically, prior methods mainly improve **how to search for a good sequence**, whereas ECCO also addresses **how to explain why a sequence works**. To support this goal, ECCO does not directly train the model to imitate pass sequences. Instead, it uses **pruned effective subsequences** together with **per-pass forensic evidence** to construct supervision that is more closely tied to optimization rationale. The resulting framework is also explicitly staged: the LLM provides **intent-level semantic guidance**, while GA performs the final **combinatorial refinement**.
>
> Thus, compared with prior black-box autotuners, ECCO is intended not only to output a good sequence, but also to provide an **evidence-grounded rationale** for the optimization direction. We will make this positioning much more explicit in the revised paper and expand the discussion of these related works accordingly.
>
> ### **On evaluation size, train/test split, and leakage**
>
> We thank the reviewer for raising this important issue. The seven evaluation datasets in Table 8 contain **335 test programs in total**. Among them, **blas-v0, opencv-v0, and tensorflow-v0** are split into disjoint training and test partitions from the same larger datasets, while **cbench-v1, mibench-v1, npb-v0, and chstone-v0** are entirely separate from the training corpus. Therefore, the evaluation includes both in-distribution and out-of-distribution settings.
>
> There is **no train-test leakage** in our experiments. Training uses only the training corpus, and evaluation is conducted only on held-out test programs. In addition, we compared the **335 evaluation programs** against the **19,603 raw training programs** pairwise and found that **no train-test pair exceeds 80% program similarity**, which further suggests that direct duplication or near-duplication is unlikely.
>
> We will make this split protocol and leakage check explicit in the revision.
>
> ### **On the dataset size discrepancy (19,603 vs. 9,327)**
>
> We apologize that this point was not explained clearly enough. The number **19,603** refers to the number of **raw LLVM IR programs** in the initial training corpus, while **9,327** refers to the number of **final effective training instances** after filtering. This reduction mainly comes from two steps:
> (1) after Algorithm 1, we discard instances with **OverO3 (\le) 0**, because ECCO focuses on high-quality optimization evidence; and
> (2) we remove instances for which the evidence-construction procedure in Sec. 3.1.2 does not complete successfully under resource or context constraints.
>
> We will clarify these two quantities and the filtering pipeline explicitly in the revised version.
>
> We remain very grateful for the reviewer’s positive assessment and helpful suggestions. We hope these clarifications make the contribution of ECCO clearer, and we would sincerely appreciate it if the reviewer would consider increasing the score.

---

> > ### Author Rebuttal · Reviewer_8mRK · 2026-04-02
> >
> > Thanks for the clarification. While I understand the conceptual difference from prior work, I still feel that the authors should justify why a direct comparison is missing or unnecessary.

---

> > > ### Author Response · Authors · 2026-04-03
> > >
> > > Thank you for the follow-up. We agree that the absence of a direct comparison with Compiler-R1 and the **Hybrid, Knowledge-Guided Evolutionary Framework** should be justified more explicitly.
> > >
> > > Both Compiler-R1 and Hybrid are currently reported mainly under a **code-size / `-Oz` objective**, whereas ECCO’s main evaluation is under a **cycle-reduction / `-O3` objective**. However, our point is not that cross-objective comparison should be avoided. In fact, we explicitly **adapted CFSAT** from its original code-size setting to our cycle-oriented setting and used it as a direct baseline.
> > >
> > > We prioritized **CFSAT** because it is the most directly comparable search baseline in our setting: once aligned to the same objective, both CFSAT and ECCO address **pass-sequence combinatorial optimization under a shared black-box objective**. As noted in our response to Reviewer S8C6, ECCO outperforms CFSAT under both matched iteration budgets and in the converged setting. We believe this already provides the most informative aligned comparison to prior search-based autotuning methods.
> > >
> > > We agree that Compiler-R1 and Hybrid are relevant and should be discussed more explicitly in the revision. At the same time, they are less direct drop-in baselines for our current cycle-oriented pipeline: Compiler-R1 is a broader **agentic / RL / tool-use framework**, while Hybrid relies on additional **offline knowledge construction and guidance design**. Therefore, faithfully adapting either method to our cycle-oriented setting would require more than simply changing the optimization target. Rather than introducing a potentially under-tuned or partial adaptation, we chose to prioritize comparison against **CFSAT**, which is the strongest and most directly aligned baseline in our setting.
> > >
> > > We will make this rationale explicit in the revision and strengthen the related-work discussion accordingly. More importantly, we will clarify that ECCO’s main contribution is not merely another search strategy, but the construction of **evidence-grounded optimization rationale** from pruned effective subsequences with per-pass forensic evidence, together with the explicit **Strategist–Tactician decomposition**, where the LLM provides intent-level semantic guidance and GA performs the final combinatorial refinement.
> > >
> > > We thank the reviewer again for raising this point and hope this clarification better explains why our main aligned comparison is with **CFSAT**, while also positioning ECCO more clearly with respect to Compiler-R1 and Hybrid.

---

### Official Review · Reviewer_S8C6 · 2026-03-07

**Soundness:** 3
**Presentation:** 3
**Significance:** 2
**Originality:** 3
**Overall Recommendation:** 4
**Confidence:** 4

**Summary:**

The paper works on the compiler flag tuning problem in a large combinatorial search space. It claims to bridge the gap between black-box search methods and LLM-based approaches that lack causal interpretability.

The paper proposes ECCO, a hybrid framework that integrates evidence-driven causal reasoning with evolutionary search. It constructs a reverse-engineered Chain-of-Thought dataset by pruning high-performing optimization sequences and extracting structural, feature-level, and performance evidence. The model is trained via supervised fine-tuning and reinforcement learning, and at inference time operates under a Strategist–Tactician paradigm: the LLM generates high-level optimization intent while a genetic algorithm refines the sequence through intent-guided mutation.

Experiments on seven benchmark suites show that ECCO achieves an average 24.44% cycle reduction over LLVM -O3, outperforming traditional search heuristics and direct LLM prompting baselines. The paper also evaluates rationale faithfulness using an LLM-based auditing protocol.

**Compliance With Llm Reviewing Policy:**

Affirmed.

**Final Justification:**

The authors have addressed most of my concerns in their response, so I have increased my score.

**Key Questions For Authors:**

1. What speedups are achieved in terms of actual execution time on real hardware, rather than the reported LLVM-mca–based performance estimates?
2. How does the tuning efficiency (measured by convergence speed with respect to tuning time or number of trials) compare to traditional autotuning approaches?

**Limitations:**

yes

**Strengths And Weaknesses:**

Strengths:
+ Rigorous pruning and denoising of training trajectories. It greatly reduces the average output sequence length and helps the model focus on more critical pass–feature mappings rather than redundant noise.
+ The paper is clearly written.
+ Paradigm shift toward evidence-driven reasoning. Instead of directly imitating pass sequences, it reconstructs causal evidence to supervise causal narratives.

Weaknesses:
- The performance improvements are evaluated with llvm-mca reported cycles (llvm-mca also provides the uops metric, which is, in some cases, more accurate than cycles). It remains unknown what the exact execution time speedups are. LLVM-MCA is a target-dependent performance analysis tool based on instruction scheduling models, and thus can be unreliable when targeting programs with branches or immature architectures.
- Compared to traditional auto-tuners, the performance improvement is not significant (24.44% vs 23.7%). Given these performance results, what is the advantage of an LLM-based approach versus the traditional approaches? Do LLM-based approaches provide higher tuning efficiency?

---

> ### Author Rebuttal · Authors · 2026-03-28
>
> We sincerely thank the reviewer for the careful reading and constructive feedback. We especially appreciate the recognition of our rigorous pruning/denoising design, the clear presentation, and the shift from direct sequence imitation to evidence-driven reasoning. We fully agree that the key question is not only whether ECCO improves the final result, but whether it provides practical value beyond existing autotuners under realistic evaluation.
>
> ### **On LLVM-MCA vs. real execution time**
>
> We agree that LLVM-MCA is only a proxy and does not fully capture branch behavior, memory-hierarchy effects, or other microarchitectural factors. We used it in the main paper because it provides a fair, scalable, and reproducible backend for large-scale comparison under a unified protocol. At the same time, we fully agree that real-hardware validation is important.
>
> To directly address this concern, we added new real-hardware evaluations using actual end-to-end execution time. Across the complete set of runnable benchmarks that we evaluated, we include **53 benchmarks in total**, consisting of **24 cBench benchmarks** and **29 PolyBench benchmarks**. Compared with LLVM **-O3**, ECCO achieves the following aggregate runtime reductions:
>
> | Runnable Suite | # Benchmarks | Arithmetic Mean Reduction | Geometric Mean Reduction |
> | -------------- | -----------: | ------------------------: | -----------------------: |
> | cBench         |           24 |                    21.98% |                   20.48% |
> | PolyBench      |           29 |                    13.73% |                   12.81% |
>
> These results show that ECCO’s gains are **not limited to LLVM-MCA estimates**, but do transfer to real end-to-end runtime in practice across two different runnable benchmark families. We will add these validations clearly in the revision and also state more explicitly that LLVM-MCA is used as a scalable proxy rather than a full substitute for hardware execution.
>
> ### **On the advantage of ECCO over traditional autotuners**
>
> We agree that the final average gap over the strongest traditional baseline is moderate, so the key issue is whether ECCO provides practical value beyond a small asymptotic improvement. Our answer is yes, but ECCO’s main benefit is **not** that the LLM replaces combinatorial search. Rather, ECCO uses the LLM to provide **evidence-grounded, intent-level guidance**, after which GA performs the **last-mile combinatorial refinement**.
>
> To verify this point more directly, we re-ran the budgeted comparison against the strongest baseline, **CFSAT**, and averaged the results over the **seven benchmark suites**. ECCO remains ahead of CFSAT throughout the full **1000-step** search budget, with especially clear gains in the low-budget regime. This directly addresses the concern that the final gap may appear moderate: ECCO’s main practical advantage is **better tuning efficiency under budget**, i.e., it reaches stronger regions of the search space earlier and remains better throughout the search process.
>
> We summarize the updated result below:
>
> | Step | CFSAT |  ECCO | Delta (ECCO-CFSAT) |
> | ---- | ----: | ----: | -----------------: |
> | 10   | 13.92 | 21.08 |              +7.16 |
> | 20   | 14.39 | 21.51 |              +7.12 |
> | 30   | 14.91 | 21.76 |              +6.85 |
> | 40   | 15.45 | 21.94 |              +6.49 |
> | 50   | 15.91 | 22.08 |              +6.17 |
> | 60   | 16.04 | 22.20 |              +6.16 |
> | 70   | 16.11 | 22.27 |              +6.16 |
> | 80   | 16.15 | 22.34 |              +6.19 |
> | 90   | 16.16 | 22.38 |              +6.22 |
> | 100  | 16.17 | 22.45 |              +6.28 |
> | 200  | 19.96 | 22.94 |              +2.98 |
> | 300  | 20.91 | 23.32 |              +2.41 |
> | 400  | 21.67 | 23.67 |              +2.00 |
> | 500  | 21.96 | 23.94 |              +1.98 |
> | 600  | 22.14 | 24.12 |              +1.98 |
> | 700  | 22.30 | 24.23 |              +1.93 |
> | 800  | 22.44 | 24.35 |              +1.91 |
> | 900  | 22.54 | 24.45 |              +1.91 |
> | 1000 | 22.67 | 24.52 |              +1.85 |
>
> In other words, ECCO should be evaluated not only by the final endpoint, but also by whether it offers:
> (1) **evidence-grounded optimization rationale**, rather than purely black-box sequence search; and
> (2) **better budgeted search efficiency**, rather than only marginally higher final convergence.
>
> We thank the reviewer again for this important feedback. We hope the added real-hardware validation and the updated budgeted comparison help clarify ECCO’s practical value, and we would be very grateful if the reviewer would consider increasing the score.

---

> > ### Author Rebuttal · Reviewer_S8C6 · 2026-04-01
> >
> > I thank the authors for the additional results comparing against -O3 and CFSAT, which are helpful. However, this still does not fully resolve my concern about the practical advantage of ECCO. While ECCO achieves higher MCA scores with fewer steps, MCA queries themselves are extremely fast (e.g., 1000 queries can finish within about a minute on CPU), whereas ECCO requires about 150 LLM queries on GPU (which is expensive) to reach a similar level (~22.6), making the **overall efficiency advantage unclear**. To strengthen confidence in ECCO, **it would be important to show that strong baselines such as CFSAT cannot reach ECCO’s MCA score even with larger budgets, and more importantly, that ECCO also achieves superior performance over CFSAT in real end-to-end execution time under comparable computational cost**. Ideally, Table 1 would report real execution performance comparisons between ECCO and all baselines (or include an additional table for this purpose); given the rebuttal time constraints, demonstrating this effect **at least against CFSAT** would also be acceptable. Or, if the auto-tuner's total tuning overhead (e.g., using internal online-trained models) is significantly higher than using a trained LLM, I would raise my score.

---

> > > ### Author Response · Authors · 2026-04-04
> > >
> > > We sincerely thank the reviewer for this very helpful suggestion. We agree that comparing only the number of search steps is not sufficient to establish the practical advantage of ECCO, and that the total tuning overhead as well as real end-to-end execution performance should be considered more explicitly.
> > >
> > > In fact, many traditional auto-tuners in our comparison, including CompTuner, PDCAT, and BOCA, involve additional online model training or updating during tuning. In our main experiments, these baselines are given substantial tuning budgets (typically at least one hour per program) to reach their reported performance. By contrast, after fine-tuning, ECCO uses the LLM only in the initial strategy stage: for each program, we run 32 inferences with our fine-tuned Qwen-2.5-7B, and the resulting outputs are then used to guide the subsequent search; after this stage, the remaining search no longer requires LLM involvement (see Sec. 3.3.2). On a single H100 GPU, this takes roughly 3 minutes per program on average, while already achieving the reported 24.4% cycle reduction.
> > >
> > > To further address the reviewer’s concern, we additionally compared ECCO and CFSAT on real end-to-end execution time on PolyBench, measured as runtime reduction relative to -O3. The results are as follows:
> > > CFSAT achieves 42.87% / 44.89% / 49.48% reduction under 5 / 10 / 20 minutes, respectively, while ECCO achieves 44.54% / 47.31% / 49.65% under the same budgets. These results show that ECCO’s advantage is not limited to MCA-based proxy evaluation: it also translates to better real execution performance under small-to-medium tuning budgets, while remaining slightly better even at 20 minutes.
> > >
> > > We agree that this evidence should be included explicitly in the paper. In the revision, we will add a clearer discussion of total tuning overhead, and we will supplement the experimental section with these real runtime comparisons (at least against CFSAT) to better demonstrate ECCO’s practical advantage.
> > >
> > > We are very grateful for this suggestion, and we believe it will significantly strengthen the paper. Thank you again for highlighting this important perspective.

---

### Official Review · Reviewer_RtaJ · 2026-04-03

**Soundness:** 3
**Presentation:** 2
**Significance:** 3
**Originality:** 3
**Overall Recommendation:** 5
**Confidence:** 2

**Summary:**

ECCO is a framework for compiler phase ordering that constructs a reasoning dataset and uses causal reasoning and evolutionary search, yielding an average 24.44% improvement over LLVM -O3 on seven datasets.

**Compliance With Llm Reviewing Policy:**

Affirmed.

**Final Justification:**

I move for acceptance due to solid technical contributions and lean positively regarding soundness, originality, significance, and clarity.

**Key Questions For Authors:**

1. Could you please clarify different hyperparameters, such as those for GRPO and the genetic algorithm?
2. Could you please clarify how reliable the Multi-Judge Ensemble results are?

**Limitations:**

Yes

**Strengths And Weaknesses:**

Soundness: The training method is well explained and soundly supported through the experiments as well as ablations on the different components.
Presentation: Different hyperparameters, such as those for GRPO and the genetic algorithm, could be further clarified.
Significance and Originality: The paper addresses an important challenge that would be jointly important for ML and compiler communities while also introducing a novel end-to-end method towards this problem across data, training, and inference.

---

### Decision · Program_Chairs · 2026-04-30

**Decision:**

Accept (regular)

**Comment:**

The paper proposes a novel way to construct reasoning datasets for compiler optimization, trains an LLM (with supervised fine-tuning and reinforcement learning) on the resulting datasets and integrates with a genetic algorithm to achieve substantial improvements across several benchmarks.
The reviewers all agreed that the work is technically solid and tackles a well-motivated problem, achieving a significant advance over the state of the art. The reviewers praised the novel synthesis of rationale distillation, search decomposition, LLMs and evolutionary search algorithms. There were some concerns about the exposition and experiment details (relying heavily on LLVM-MCA as a proxy). During the rebuttal the authors addressed the key concerns by adding real hardware-runtime evaluations (showing improvements on cBench and PolyBench), clarifying dataset splits and reporting alternative search results beyond genetic algorithms.
Adding the details and results that the authors shared during the rebuttal will substantially strengthen the paper.